# Active Commuting and Healthy Behavior among Adolescents in Neighborhoods with Varying Socioeconomic Status: The NESLA Study

**DOI:** 10.3390/ijerph19073784

**Published:** 2022-03-22

**Authors:** Benti Geleta Buli, Annika Tillander, Terence Fell, Katarina Bälter

**Affiliations:** 1Division of Public Health Sciences, Mälardalen University, 722 20 Västerås, Sweden; katarina.balter@mdu.se; 2Department of Computer and Information Science, Linköping University, 581 83 Linköping, Sweden; annika.tillander@liu.se; 3Division of Economics and Political Science, Mälardalen University, 722 20 Västerås, Sweden; terence.fell@mdu.se; 4Department of Medical Epidemiology and Biostatistics, Karolinska Institutet, 171 77 Stockholm, Sweden

**Keywords:** active transport, school, lifestyle, health, environment

## Abstract

(1) Background: The World Health Organization recommends active commuting as a source of physical activity. Active commuting is determined by various factors, including the socioeconomic status (SES) of families and neighborhoods, distance to schools, perceived neighborhood safety, lifestyles and availability of walkways and biking paths. This study aimed to assess factors associated with modes of transportation to and from school among adolescents aged 16–19 living in a middle-sized city in Sweden. (2) Method: Three hundred and fourteen students, of whom 55% were females, from schools in the city of Västerås participated in the study. Printed as well as web-based self-administered questionnaires were used to collect the data. (3) Results: Adolescents living in high SES neighborhoods were 80% more likely to bike or walk to school (OR = 1.80; CI: 1.01, 3.20) than adolescents living in low SES neighborhoods. Furthermore, active commuting was associated with higher consumption of fruits and vegetables (OR = 1.77; CI: 1.05, 2.97) and less consumption of junk foods (OR = 0.43; CI: 0.26, 0.71), as compared to passive commuting. (4) Conclusions: Active commuting is a cost-effective and sustainable source of regular physical activity and should be encouraged at a societal level.

## 1. Introduction

The World Health Organization (WHO) recommends active commuting, i.e., walking or biking, to and from school or work as a regular source of physical activity [1]. Studies have shown that adolescents who actively commuted to or from school reported higher total physical activity levels than those who traveled by motorized transport, known as passive commuters [2,3,4,5,6]. Active commuting has also been associated with increased likelihood of participating in various physical activities, such as school sports, which in turn increased overall physical activity levels [7].

Physical activity is an important lifestyle factor that helps to attain physical fitness by enhancing cardiovascular and musculoskeletal fitness, and by regulating body composition and metabolism [8]. Studies have shown that among adolescents, active commuting, especially biking, was associated with lower body mass index (BMI) and lower risk of obesity compared to passive commuting [9,10]. However, many active commuting students live too close to schools to meet the minimum health recommendation of at least 60 min per day of moderate-to-vigorous intensity physical activity from active transports only, where walking is considered an important complementary contribution to total daily physical activity [4,11,12,13,14,15].

Moreover, active commuting has positive environmental dimensions as it does not contribute to greenhouse gas emissions, traffic congestion, air pollution or noise. It is among the suggested interventions in a study conducted among young people in Nordic regions in which 89% were concerned about climate changes and 93% thought that a sustainable lifestyle is important [16,17]. Also, active commuting contributes to economic benefits to families and communities in terms of expenses, related but not limited to, reduced fuel usage, repair and maintenance costs, and reduced health care as well as road safety costs [18,19,20]. Apart from health, economic, and environmental aspects, a study in Canada among girls aged 7–15 years highlighted that active commuting may be fun, social, and a sign of independence among young people [21].

Previous studies have shown that social and built environments including availability of walkways, biking paths, parks, and perceived neighborhood safety, as well as social interaction among adolescents who live together in a neighborhood were found to be associated with active commuting [22,23,24,25,26]. Moreover, shorter distance to school, urban setting and neighborhood density were reported to be positively associated with active commuting [27,28,29].

Studies on active commuting among adolescents aged 16–19 years are sparse [30,31,32]. Also, differences in infrastructure, climate and seasonal variations as well as socioeconomic factors influence active commuting worldwide, making comparisons difficult. Compared to high SES families and neighborhoods., low SES of families and neighborhoods were found to be positively associated with active commuting among children younger than 18 years in the U.S. [33] as well as in a study of children aged 12–17 years in Brazil [34]. A study in Spanish urban schools among adolescents aged 13–18.5 years showed a higher rate of active commuting for girls than for boys among those attending public and private schools [12]. In contrast, a study in Australia among children aged 10–14 years found higher rates of active commuting for boys than for girls [35].

Studies among children of different age groups from the U.S. (5–18 years), Canada (11–15 years), Australia (5–14 years), China (6–18 years), Brazil (7–15 years) and Switzerland (6–14 years) showed that there has been a substantial shift towards motorized modes of transportation over time [36,37,38,39,40,41]. Similarly, studies in Sweden indicate a declining trend in active commuting to and from school. A study from Sweden found that 68% of children and adolescents aged 9–10 and 15–16 years walked or biked in 1999 [3], whereas a national survey, the Swedish version of ‘Health Behavior in School-aged Children’ (HBSC) from 2006, reported that 63% of children aged 11, 13 and 15 years walked or biked to school [42]. The latter study also showed that the active commuting behavior decreased by age, where 76% of children aged 11 years, 62% of those aged 13, and 50% of those aged 15 years walked or biked to and from school, while no differences were reported when considering gender or ethnic background. 

Previous studies have found that active commuting is linked to other healthy behaviors. For example, frequent consumption of fruits and vegetables was more common among active commuters, whereas junk foods i.e., sweets, chips, hamburgers, pizzas and/or hotdogs, were eaten less often, compared to students who were non-active commuters [43,44,45,46]. The aim of this study was to assess the association between active commuting and aspects of healthy behaviors among adolescents aged 16–19 years living in neighborhoods with varying SES in the middle-sized city of Västerås in Sweden. Conducive to the study, the city has 380 km of walking and biking paths that cover the whole city [47], as shown in Figure 1.

## 2. Materials and Methods

### 2.1. Study Design and Sample

We used data from the Neighborhood, Sustainable Lifestyle and Health among Adolescents (NESLA) project. NESLA is grounded in the United Nation’s sustainability goals and aims at studying lifestyle factors and health among adolescents aged 16–19 years old living in the city of Västerås (*n* = 150,000) in Sweden. The study was approved by the Ethical Review Board of Uppsala.

During the fall of 2017, we contacted the principals of 21 upper secondary schools (i.e., high schools) in Västerås, and, as shown in Figure 1, six of them responded that their schools were interested in participating. We visited 21 classes from these schools, and the students were asked to fill out a printed questionnaire. After the students had finished the questionnaire, we gave a short presentation of Agenda 2030. In addition, schools from the area that signed up for a scientific event (Science@mdu) at our home university, i.e., Mälardalen University, during the fall of 2017 were also given the opportunity to fill out a web-based version of the same questionnaire before the event. These were adolescents from the same schools as well as from other schools in the area. During the event, they were given a presentation about sustainable lifestyle and health. Both the printed and web-based questionnaire included questions about which neighborhood they lived in, features of their neighborhood, healthy behavior (food habits, physical activity), environmental awareness, school, and their families’ SES. 

A total of 554 students completed the questionnaire; 397 (72%) filled out the printed and 157 (28%) the web-based questionnaire.Of these, 422 (76%) were from schools in Västerås and the rest from schools in the neighboring city of Eskilstuna, where our university has another campus and where one of the science events took place. Since many students commute to schools in Västerås from other smaller towns in the county, at a distance that is too far for active commuting, we only included students living in the city of Västerås for the present analyses. This step reduced the sample size to 333. In further cleaning of the data, we found out that the main variable from which we coded the outcome variable “active commuting” had 19 missing values that prompted dropping them. Therefore, the final sample size was 314. Thus, inclusion criteria for the present study were adolescents living in the City of Västerås and enrolled at an upper secondary school in Västerås during the fall of 2017 and that filled out the questionnaire, either the printed or the web-based version. Consequently, we excluded adolescents that lived outside of Västerås and therefore were unable to practice active commuting as well as adolescents enrolled in an upper secondary schools in the City of Eskilstuna.

### 2.2. Active Commuting to and from School

To assess mode of transport to and from school, as part of everyday physical activity, the following question was posed: “How do you usually get to and from school?” The students were asked to select one of the following options: public transportation (bus or train), biking, walking, moped, or car. Students who reported that they biked or walked to and from school were categorized as “active commuters”, and students reporting that they used any motorized mode of transport (public transport, moped or car) were categorized as “passive commuters”. 

### 2.3. Socioeconomic Characteristics

We used two indicators to assess SES; one was a subjective indicator and the other one was objective. The subjective SES indicator used the ladder metaphor to help people position themselves somewhere relative to their fellow society members [48]. To measure subjective SES, we presented the following question to the students: “Imagine society as a ladder. Families with the least money are at the bottom of the ladder, while at the top are those with the most money. If you think about your family’s wealth compared to the society at large, where would you place your family on the scale?”. The students then self-reported their families’ SES using a scale of seven levels, in which the lowest levels denote the lowest wealth, and seven denotes the highest level of wealth. We then placed each family, according to economic status, in three evenly large groups by breaking the scale of seven levels into three categories. The lowest three levels of the ladder were categorized as “Low”, the fourth as “Middle” and the top three levels as “High”. This means that the subjective SES status indicator provided information about how the adolescent identified her/his family as belonging to either low, middle or high SES in the society.

To assess the objective SES of the 22 different neighborhoods in Västerås, four variables were obtained from Statistics Sweden online database specific to the city of Västerås [49] to create a rank index ($I_{\mathrm{SES}}$). These four variables were the proportion of persons with higher education (HE), mean disposable income (DI), proportion of single parent households (SP), and proportion of people who were unemployed (UE). Variables HE and DI were ranked from lowest to highest value, where the highest possible rank was 22, while the variables SP and UE were reciprocally ranked, with the lowest value ranked 22. The ranks of the four variables were then added together, and the sum was divided by 88, which is the highest possible rank sum for each neighborhood. This index was generated to indicate where each neighborhood is positioned with regard to SES compared to the other neighborhoods in Västerås. Higher index value indicates higher SES rank of that neighborhood in comparison to other neighborhoods in the city.
$$I_{\mathrm{SES}}=\left(\frac{{\mathrm{HE}}_{\mathrm{rank}}+{\mathrm{DI}}_{\mathrm{rank}}+{\mathrm{SP}}_{\mathrm{rev}-\mathrm{rank}}+{\mathrm{UE}}_{\mathrm{rev}-\mathrm{rank}}}{88}\right)\cdot 100$$

The neighborhood SES index was then divided into tertiles for the ease of analysis and interpretation of the results, see Table 1. 

The two SES indices reflect different aspects of the individual’s life. The subjective index is an indication of the SES of the individual family, and the objective index is an indication of the status of the neighborhood. The Spearman correlation coefficient between the two was 0.17. The poor correlation may be due to the fact that they measure two different aspects of SES, in which the subjective one is relative position in the society [48] and the other is an objectively verifiable measure of wealth [49]. 

### 2.4. Lifestyle Factors

Students self-reported their fruit and vegetables consumption as well as eating junk food. They were asked to select one intake frequency out of “more than once a day”, “daily”, “a few times a week”, “once a week” or “seldom or never”. Those who consume fruits and vegetables daily or more than once a day were categorized as “frequent consumers”, while the rest were categorized as “non-frequent consumers”. Consumption of junk food was based on two different questions. The first was about consumption of candy, chips and other sweets, and the other question was about how often they eat at fast food restaurants, pizzerias or hot dog stands/kiosks. The total consumption frequency from those two questions was summed up, and those who reported eating junk foods more than once a day, daily and a few times a week were categorized as “frequent consumers”, while the rest were categorized as “non-frequent consumers”.

Organized physical activities, i.e., exercise, were assessed by asking about membership in any sport club or organization. Students who reported that they were members in any sport club or organization, including the football team, were categorized as “member in sport club”, while the others were categorized as “non-members in sport clubs”.

### 2.5. Structure and Service 

The availability of suitable public transportation was assessed using the following question. “Are there good public transportations in the neighborhoods where you live?” and the responses were coded as “Yes”or “No”. The students were asked about availability of well illuminated walkways and bike paths in their neighborhoods. The question presented to the students was: “Are the walking and bike routes in the area where you live illuminated?”, and the responses were “Yes” or “No”.

### 2.6. Statistical Analysis 

The study employed logistic regression to analyze the association between active commuting, and neighborhood SES, family economic status, gender, self-reported consumption of fruits and vegetables and junk foods, illumination of walkways and biking paths, membership in any sport club, concern about environmental impact, and belief that one’s own lifestyle can influence the environment, respectively. To build the model, we first conducted univariate analysis to the variables that were initially selected from the NESLA dataset based on information in literature on their association with the dependent variable, active commuting. This was done in order to identify the variables that were at least moderately associated with the dependent variable. We then added one covariate at a time, starting with the main explanatory variable, the neighborhood SES, until we included all the variables listed above. We then removed from the model the explanatory variables that did not have significant association with the outcome variable. The final model contains active commuting, the outcome variable, neighbor SES, illumination of walkways or biking paths, consumption of fruits and vegetables, and consumption of junk food. At the end, we used the Hosmer-Lemeshow test and Nagelkerke’s R^2^ to test the goodness-of-fit of the final model.

### 2.7. Construction of Map

A map has been constructed in QGIS 3.10.2 to display three main indicators: neighborhood SES index divided into tertiles, walking and biking networks and location of the six schools that we visited, and adolescents filled out a printed questionnaire. We received three shapefiles from the City of Västrås for this purpose: (1) a polygon that shows administrative borders of the City of Västerås and its neighborhoods, (2) a point shapefile that shows location of schools in the city and (3) a line shapefile that shows walking and biking networks in the city. We used the administrative shapefile to stratify the neighborhoods as low, middle or high based on SES tertiles. The tertiles were calculated using information shown in Table 1. We selected the surveyed schools from the list of schools in the attributable table and displayed in the map. The walking and biking networks’ layer was overlaid without any change except coloring.

## 3. Results

Baseline characteristics of the participants are shown in Table 2. The study was comprised of 174 female students (55%), 127 male students (40%) and eight (3%) who did not identify their gender, with five (2%) missing values. The mean age of the participants was 17.3 years, and in total, 32% (35% for boys and 29% for girls) of the students reported to have either biked or walked to and from school. In the whole group, 8% commuted to school by walking, 24% by biking, 61% by public transportation, 2% by moped and 5% by car. Thirty-five percent of the students were members of a sport club, 57% reported eating fruits and vegetables daily or more often, and 60% reported eating junk food a few times per week or more often. The majority of the participants (60%) perceived their families as belonging to the middle or high disposable income category, which was quite comparable to the finding from the objective index in which 63% lived in a neighborhood with middle or high SES.

The associations between active commuting and aspects of healthy behavior among adolescents living in neighborhoods with varying SES are presented in Table 3. Active commuting to and from school was 80% more common among students from neighborhoods with high SES than those from neighborhoods with low SES (OR = 1.80; CI: 1.01–3.20). Furthermore, active commuting was 50% less common among adolescents from neighborhoods with middle SES than among those in low SES neighborhoods. In contrast, subjectively reported individual family economic status was not associated with active commuting to and from school.

Positive association was observed between active commuting to and from school and consumption of fruits and vegetables. Active commuting was 77% more common among adolescents who reported consumption of fruits and vegetables daily or more than once a day than those who consumed fruits and vegetables a few days in a week or less (OR = 1.77; CI: 1.04–2.97). On the other hand, a negative association was found between active commuting and consumption of junk food. Active commuting to and from school was 57% less common among students who reported consuming junk food more than once in a week than those who reported seldom or non-consumption of junk food (OR = 0.42; CI: 0.25–0.71).

The results also suggested that active commuting to and from school was almost three times more common among adolescents from the neighborhoods where the walk and bike paths are illuminated than among those from neighborhoods where the paths are not illuminated (OR = 2.72; CI: 1.00–7.46). Membership in any sport club, availability of good public transport, concerns about environmental impact, and belief that personal lifestyle can influence the environment were not associated with active commuting. Effect sizes of individual variables in this model are presented in Appendix A.

The strata of neighborhood’s objective SES, locations of surveyed schools and walk and bike paths in the city of Västerås are shown in Figure 1. Five of the 6 schools are located at the city center, and the longest possible commute to any of them is about 10 km, but for a majority of the participants, it is less than that.

## 4. Discussion

The proportion of active commuters among adolescents in the city of Västerås was 32%, and these mostly lived in high SES neighborhoods. Moreover, active commuting was associated with other healthy behaviors, including more frequent consumption of fruits and vegetables and less frequent consumption of junk food.

The proportion of active commuters in the present study is lower than what was reported in a study among a national sample of Swedish children aged 11–15 years in 2006, in which active commuting was at 63%. Similar to this study, the 2006 study used self-report data [42]. Even though decline in active commuting has been reported worldwide [36,37,38,39,40,41], this study suggests that there was a rapid decrease in active commuting in Sweden, in which it declined from 63% in 2006 to 32% in this small-scale study. This is a major decrease, even if it has transpired over two decades. Ham, et al. [40] estimated the decline in active transport in the U.S. among students aged 5 to 18 years was 9% per decade. Moreover, there was a small, but not statistically significant, difference in the proportion of active commuters among boys (35%) and girls (29%) in the present study, which is in line with a study from Australia, where active commuting was more prevalent among boys than among girls aged 10–14 years [35]. This finding, however, contrasts with findings in a study among adolescents aged 13–18.5 years from Spain, where active commuting was more common among the girls than among the boys [12], indicating that there are local and cultural differences.

This study showed that active commuting to and from school differed by the SES of the neighborhood. Active commuting was 80% more common among adolescents from neighborhoods with high SES according to an objective index than those from neighborhoods with low SES. Studies elsewhere [22,23,24,25,26] attribute this difference to differences in the availability and quality of built environments such as walkways, biking paths, parks, and illumination of the walkways and biking paths that in turn could vary depending on the SES of the neighborhoods. The City of Västerås has an extensive network of walk and bike paths, as seen in Figure 1, and are well developed in both neighborhoods with low and high SES. However, it is beyond the scope of our study to assess the quality of the paths in different neighborhoods. It is worth noting, however, that most schools included in this study are in high SES neighborhoods that may require students from other SES category neighborhoods to take alternative, especially motorized, modes of transportation. However, most of the schools are situated in the city center, which is a commercial area with a high density of stores and restaurants but is situated geographically near neighborhoods with low SES status. Nevertheless, this underscores the importance of data on distance between home and school, which this study did not cover.

In line with studies elsewhere [26,50], this study found that illumination of walkways and biking paths was associated with increased likelihood of active commuting to and from school. Illumination improves visibility of fellow path users, speed breakers, turns and crossings and helps reduce risks of accidents and injuries, hence ensuring personal safety. It also plays a greater role in ensuring the personal security of pedestrians and bikers [51]. This means that the sense of ensured safety by pedestrians and bikers would in turn become a major motivation for walking and biking to and from school. On the other hand, another study revealed that walking and biking behavior is significantly impacted by the presence or absence of daylight that could in turn greatly vary with season [52]. Data from national meteorological agencies show that European cities have varied annual average number of sunny hours, and Swedish cities have fewer sunny hours than most cities in central and southern Europe [53]. This may have implications regarding differences in the proportion of active commuters between studies. In this study, we do not have seasonal data, but all the data were collected during the same season, i.e., the fall, and data regarding season should be included in future studies.

This study has shed light on the association between dietary behaviors and mode of transport to and from school. Adolescents that reported consuming junk food more than once a week were 57% less likely to actively commute to and from school than those who reported consuming junk foods only once per week or never. This finding is consistent with reports from the U.S., Saudi Arabia and Nepal, where consumption of junk food was negatively associated with physical activity, including active commuting to and from school [43,44,45].

Frequent consumption of fruits and vegetables was the other healthy behavior that was found to be associated with active commuting to and from school. Active commuters reported eating fruits and vegetables more often than non-active commuters, which may be explained by the theory of clustering of behaviors in which an execution of one healthy behavior would lead one to adopt and execute another [54]. It can also be implied that families in high SES neighborhoods can afford to maintain their children’s proper diet and purchase bikes.

The results highlight the importance of future school-based interventions to promote active commuting. Ideally, the intervention should be incorporated into the regular teaching curriculum, in subjects such as biology and physical education, but also social science, since active commuting is linked to several of the global sustainability goals, in particular goals 3 (good health and well-being), 11 (sustainable cities and communities) and 13 (climate actions) [55]. Also, it is in line with the WHO guidelines on physical activity and sedentary behavior, in which the recommendation is at least 60 min of moderate intensive physical activity per day for adolescents up to age 17 [56]. A school-based intervention has the benefit of reaching all adolescents, even those from neighborhoods with low SES. To facilitate the transition to more active transports, practical support should also be available, including regular servicing of bikes and safe storage of bikes at the schools, as well as information about where to find affordable secondhand bikes.

## 5. Strengths and Limitations

A strength of the present study is that students came from many different neighborhoods in the city and attended different schools; thus, the data represent a variety of commuter routes. Another strength is that we have included both subjective and objective indices of SES, which reflect two different aspects of the individual’s life. The subjective index is an indication of the SES of the individual family, and the objective index is an indication of the status of the neighborhood.

On the other hand, use of self-reported data on lifestyle factors could be susceptible to social desirability bias. The questions asked in this study, however, were direct and had no intention of comparing individuals based on their responses [57]. The study did not take into account seasonal variations, although all data collection took place during the same season (fall). In a country like Sweden with four seasons, this influences people’s behavior. However, the City of Västerås has 380 km of walking and biking paths that cover the whole city [47], and a large proportion of the paths are heated in the winter, making them free from snow and ice, encouraging and allowing residents to stay active. Another limitation of this study is that we did not ask students about the distance in kilometers between home and school, which might influence their willingness to walk and bike. However, the longest possible distance, i.e., the neighborhood furthest from any of the schools is about 10 km, a commute that corresponds to up to 50 min of biking [58], but a majority of the students had a much shorter commute. Finally, the fact that most schools included in this study can be found in high SES inner-city neighborhoods can at first glance be interpreted as another limitation. However, Västerås’ inner city is an area with mostly stores, restaurants and workplaces, and has fewer households than neighborhoods further away from the city center. Therefore, the reason these schools are located in this part of Västerås is not because a lot of kids live there, but because it is central and easy to reach by bus for kids from many neighbourhoods with varying SES. This is in line with Sweden’s ongoing efforts to implement a variety of strategies that prevent or reduce segregation in its school system [59]. Consequently, the inner city’s high SES has little effect on the results of this study as these schools accept pupils from neighborhoods all over the city as well as surrounding areas. 

## 6. Conclusions

Active commuting is a cost-effective and sustainable source of regular physical activity [1,20] and should be encouraged at a societal level. Here we show that the SES of the neighborhood where adolescents live, combined with the presence of illuminated walking and biking paths, was associated with active commuting among adolescents aged 16–19 years. Moreover, active commuting was linked to other healthy behaviors, including more frequent daily consumption of fruits and vegetables and less frequent consumption of junk food. This study assessed active commuting among adolescents during the time period before COVID-19 pandemic. We know from other studies that COVID-19 is associated with decreased physical activity [60]. During part of the pandemic, upper secondary schools practiced online learning; thus, adolescents did not have to commute at all. However, when teaching was done in schools, it is possible that some may have chosen to avoid crowded buses and practiced active commuting instead. Therefore, further studies may be required to investigate active commuting practice in relation to the COVID-19 pandemic.

## Figures and Tables

**Figure 1 ijerph-19-03784-f001:**
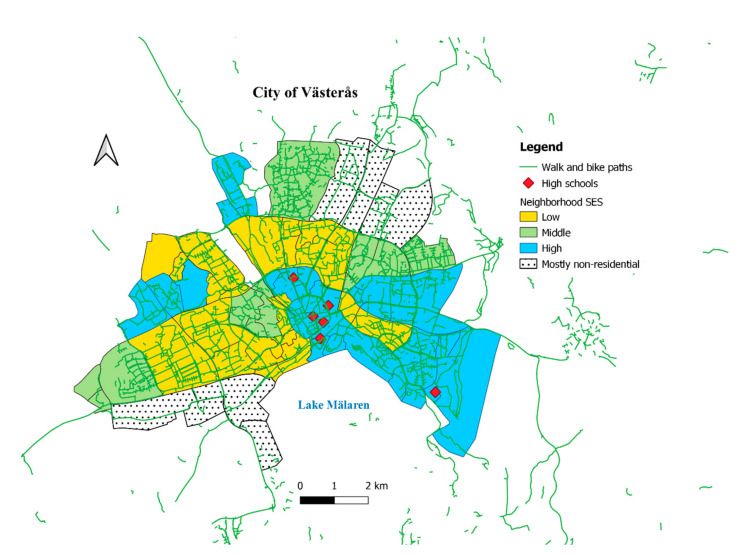
Socioeconomic status of neighborhoods, walk and bike paths in the municipality of Västerås and location of upper secondary schools included in the NESLA Study.

**Table 1 ijerph-19-03784-t001:** Description of the process of indexing the socioeconomic status of the 22 neighborhoods in the city of Västerås. Note that not all 22 neighborhoods are shown, but rather an example.

Neighborhood	^1^ DI-Rank Max = 22 Min = 1	^2^ HE-Rank Max = 22 Min = 1	^3^ SP-Rank Max = 22 Min = 1	^4^ UE-Rank Max = 22 Min = 1	Rank Sums Max = 88 Min = 4	SES Index (Min = 4.5, Max = 100)	SES Index Tertiles (T) T1: ≤ 33% T2: 33–66% T3: ≥66%
Neighborhood-1	Rank = 6	Rank = 15	Rank = 5	Rank = 19	Sum = 45	51	2
Neighborhood-2	Rank = 2	Rank = 1	Rank = 9	Rank = 21	Sum = 33	38	2
Neighborhood-16	Rank = 12	Rank = 22	Rank = 15	Rank = 22	Sum = 71	81	3
Neighborhood-22	Rank = 3	Rank = 5	Rank = 8	Rank = 20	Sum = 36	41	2

^1^ DI = mean disposable income; ^2^ HE = higher education; ^3^ SP = proportion of single parent house-holds; and ^4^ UE = proportion of people who were unemployed.

**Table 2 ijerph-19-03784-t002:** Characteristics of adolescents in the NESLA study divided by active and passive commuters (*n* = 314).

Variables		Passive n ^1^ (%)	Active n ^1^ (%)	Chi-Square (*p*-Value)
Gender	Female	124 (71)	50 (29)	4.91 (0.09)
	Male	82 (65)	45 (35)	
	Identified as neither	3 (38)	5 (63)	
Age (years)	Mean	17.32	17.29	
	SD	0.76	0.74	
Modes of transportation to and from school:	Walking	-	26 (8)	
	Biking	-	75 (24)	
	Public transportation	191 (61)	-	
	Moped	5 (2)	-	
	Car	17 (5)	-	
Are walking or biking ways well illuminated in neighborhood where you live?	No	32 (89)	4 (11)	6.69 (0.01)
	Yes	181 (65)	96 (35)	
Are there good public transport options in the neighborhood where you live?	No	19 (70)	8 (30)	0.07 (0.80)
	Yes	193 (68)	91 (32)	
Are you concerned about environmental impact?	No	53 (73)	8 (20)	0.99 (0.32)
	Yes	160 (66)	81 (34)	
Do you think that you can influence environment through lifestyle?	No	36 (78)	10 (22)	2.69 (0.10)
	Yes	177 (66)	91 (34)	
How often do you eat fruits and vegetables in a week?	A few times a week	101 (75)	34 (25)	5.29 (0.02)
	Daily or more often	112 (63)	67 (37)	
How often do you eat junk foods in a week?	Once per week or less	70 (56)	54 (44)	11.96 (<0.001)
	More than once a week	142 (75)	47 (25)	
Are you a member of any sports club or organization?	No	135 (68)	64 (32)	0.002 (0.96)
	Yes	75 (68)	36 (32)	
Where do you place your family on an economic scale?	Low	46 (66)	24 (34)	0.676 (0.71)
	Middle	84 (71)	35 (29)	
	High	46 836)	42 (34)	
Objective index of the SES of their neighborhood ^2^	Low	80 (68)	37 (32)	17.02 (<0.001)
	Middle	79 (81)	18 (19)	
	High	54 (54)	46 (46)	

^1^ Missing data for the variable when the total number of participants do not add up to 314. ^2^ Objective index of the socioeconomic status of the neighborhood is based on official statistics of the neighborhoods where four variables were used to construct the index. These four variables were the proportion of persons with higher education (HE), mean disposable income (DI), proportion of single parent households (SP) and proportion of people that were unemployed (UE).

**Table 3 ijerph-19-03784-t003:** Logistic regression results of association of socioeconomic and lifestyle factors associated with active commuting to and from school.

Variable	Category	Referent category	OR ^1^	95% CI ^2^
Neighborhood SES	Middle	Low	0.50 *	0.26–0.97
Neighborhood SES	High	Low	1.80 *	1.01–3.20
Walk or biking paths	Illuminated	Not illuminated	2.73 *	1.00–7.46
Fruit and vegetable intake	Daily or more often	A few times a week or less	1.77 *	1.05–2.97
Junk food intake	More than once a week	Once a week or less	0.43 **	0.26–0.71

** *p* < 0.01, * *p* < 0.05; Nagelkerke R^2^ = 0.162; Hosmer and Lemeshow test *p* = 0.46; ^1^ OR = odds ratio; ^2^ CI = confidence interval.

## Data Availability

The data presented in this study are available on request from the corresponding author.

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
