# Peer review of "Active Commuting and Healthy Behavior among Adolescents in Neighborhoods with Varying Socioeconomic Status: The NESLA Study"

_ijerph, 2022, doi:10.3390/ijerph19073784_

Round 1

Reviewer 1 Report

I accept the Authors' answers.

Reviewer 2 Report

The revised version fully satisfies me.

This manuscript is a resubmission of an earlier submission. The following is a list of the peer review reports and author responses from that submission.

Round 1

Reviewer 1 Report

1 - this is a very important issue, considering the risks of a sedentary lifestyle.
2 - the idea of SES seems to me to be not completely defined.
It suggested a better characterization of the concept. This aspect becomes even more necessary when it is one of the research objectives.
3 - in methods it was necessary to unequivocally place inclusion and exclusion criteria.
4 - in terms of Lifestyle factors, it suggested a better characterization of the different types of food consumption, as well as the type of physical activity (not to be confused, physical activity, with physical exercise and with sport).
5 - suggested that the results should not be discussed in any way.
6 - would it be possible to get around this limitation .... how many km per day?

Reviewer 2 Report

The authors present interesting work based on data from a survey of adolescents in Västerås, Sweden. They investigated how different factors, including socioeconomic status, influence active (walking or cycling) and passive (driving) school attendance. There are some important points to be made before publication. However, a few important points should be addressed before considering it for publication.

  1. There is extensive literature on the subject, as the authors show in the introduction and discussion. What new findings are presented in the manuscript that has not been previously published?
  2. The survey was conducted in 2017. How might the results presented here be affected by the Covid-19 outbreak?
  3. In subsection 2.3, the family’s economic status was determined by self-report, while socio-economic status was determined using a rank index. Is there literature on these methods and if so, it would be worthwhile to provide references for them.
  4. The economic status of the family was measured on a 7-level scale. What justifies the middle value being based on one level (level 4) while the extreme values are based on 3 levels (low: 1-3 and high: 5-7)?
  5. Where did the data for HE, DI, SP, and UE use to calculate SES come from?
  6. What could be the reason for the poor correlation (0.17) between the self-reported family and the calculated environmental socioeconomic index?
  7. In the case of Table 2, it would be useful to use a statistical test to compare the passive and active groups and to indicate the result (p-value) in the table.
  8. For Table 2, the distribution of cases is shown by row. It should be by column for ease of comparison.
  9. The indicated final sample number is 314. However, the sample number is not uniformly 314 for Table 2. What is the reason for this? And for logistic regression analysis, what is the real sample number?
  10. There are several factors in Table 2 for which data are available. Why are the covariates shown in Table 3 included in the logistic regression model? Why not use gender and age as covariates? It would be worth testing all factories and leaving those that show a correlation with the active-passive group.
  11. Please indicate the p-values for Table 3.
  12. Figure 1 shows that the 6 schools included in the study are all located in areas with high SES. No school located in a low SES area was included in the study. This should be mentioned as a limitation.
  13. Based on the results, what intervention options could the authors recommend?
  14. Why did the authors select the "Not applicable" option for the data availability statement?

Overall, all the findings in this manuscript are based on a single descriptive table (Table 2.) and the results of a logistic regression analysis. I suggest a more complex analysis, as there is a large amount of data available that are listed in detail but not applied.

Reviewer 3 Report

This is well written study but I think that presented results by the authors are not properly analysed. Table 3 presents the results of logistic regression and two measure of goodness of fit of the model indicated that there were not significant differences between observed and predicted variables. This is some how simplification in presenting results and the Authors would be willing to add some more statistics indicating the significance of the effects of each factor, for example Wald's chi-square with a p level. It would be much more informative of a potential power of each analysed factor.